

# An effective and comprehensible method to detect and evaluate retinal damage due to diabetes complications

Quang Toan Dao[1], Hoang Quan Trinh[2] and Viet Anh Nguyen[1]

[1] Institute of Information Technology, Vietnam Academy of Science and Technology, Hanoi, Vietnam
[2] Vietnam Space Center, Vietnam Academy of Science and Technology, Hanoi, Vietnam

## ABSTRACT

The leading cause of vision loss globally is diabetic retinopathy. Researchers are making great efforts to automatically detect and diagnose correctly diabetic retinopathy. Diabetic retinopathy includes five stages: no diabetic retinopathy, mild diabetic retinopathy, moderate diabetic retinopathy, severe diabetic retinopathy and proliferative diabetic retinopathy. Recent studies have offered several multi-tasking deep learning models to detect and assess the level of diabetic retinopathy. However, the explanation for the assessment of disease severity of these models is limited, and only stops at showing lesions through images. These studies have not explained on what basis the appraisal of disease severity is based. In this article, we present a system for assessing and interpreting the five stages of diabetic retinopathy. The proposed system is built from internal models including a deep learning model that detects lesions and an explanatory model that assesses disease stage. The deep learning model that detects lesions uses the Mask R-CNN deep learning network to specify the location and shape of the lesion and classify the lesion types. This model is a combination of two networks: one used to detect hemorrhagic and exudative lesions, and one used to detect vascular lesions like aneurysm and proliferation. The explanatory model appraises disease severity based on the severity of each type of lesion and the association between types. The severity of the disease will be decided by the model based on the number of lesions, the density and the area of the lesions. The experimental results on real-world datasets show that our proposed method achieves high accuracy of assessing five stages of diabetic retinopathy comparable to existing state-of-the-art methods and is capable of explaining the causes of disease severity.

## INTRODUCTION

Diabetic retinopathy (DR), a complication of diabetes, can cause damage to the retina, the crucial part of the eye. Specifically, the damage caused to the retina by diabetes is known as "diabetic retinopathy". Diabetic retinopathy is a common complication of diabetes that occurs when blood sugar levels are poorly controlled over a long period of time. The negative effects of high blood sugar on the tiny blood vessels (microvasculature) in the

Corresponding author
Viet Anh Nguyen, anhnv@ioit.ac.vn

retina lead to abnormal changes in the retina. It is one of the most frequent diseases in the elderly, damaging the retina by diabetes, and one of the leading causes of blindness (*Trivino et al., 2018*; *Jonas et al., 2013*; *Shenavarmasouleh & Arabnia, 2021*; *Vincent et al., 2010*). The longer a person has diabetes, the higher the risk of developing diabetic retinopathy. According to statistics, diabetic retinopathy accounts for 12% of all new blindness cases, each year in the United States. Diabetic retinopathy is also the leading cause of blindness for patients from 20 to 64 years old, especially the elderly (*Li et al., 2020*; *Wang et al., 2021*; *Lakshminarayanan et al., 2021*). If not detected and treated promptly, it will cause severe damage to the fundus such as macular edema, vitreous hemorrhage, retinal hemorrhage, *etc.* leading to blindness (*Ding & Wong, 2012*; *Stevens et al., 2013*).

Diabetic retinopathy causes a number of significant conditions. Background retinopathy's symptoms include forms of the retinal capillary aneurysm, slight bleeding, stagnation of secretions in the retina, and retinal edema. The macular disease has forms of macular edema, cyst formation, or ischemic injury. The pre-proliferative disease forms caused by the abnormal blood supply to the retina, leading to ischemic lesions, hemorrhages, exudates, and retinal edema. Proliferative pathology has forms caused by the proliferation of abnormal neovascularization, causing continuous recurrent bleeding, causing organization and pulling of retinal fluid. The consequences are severe damage to the retina, and tear or detachment of the retina leading to blindness.

These pathologies are represented by the types of lesions that can be observed on fundus images. Exudate includes hard exudates and soft exudates. Hard exudates are caused by the rupture of retinal blood vessels, composed of blood fluid, lipids, and small particles. The discharge is yellowish-white with clear margins, which makes the retina thicken markedly. Soft exudate, also known as cottony discharge/cotton spot, is an edema of the nerve fibers caused by capillary ischemia in the nerve fiber layers. The shape is smooth white spots with indistinct margins, usually located in about three disc diameters where the nerve fiber layer is thickest and absent in the center of the macula because this area is supplied with blood by the choroidal system. Hemorrhage includes dot and spotted hemorrhages and flame-shaped hemorrhages. Dot and spotted hemorrhages are small, circular hemorrhages originating in the deep anterior venous capillaries. They have small round shapes because they are located in the inner nucleus and outer plexus layer, and are therefore limited by the surrounding structures. Flame-shaped hemorrhages are superficial retinal hemorrhages usually found in the nerve fiber layer, especially near the optic disc. Because they are located in the layer of striped nerve fibers, they have a flame shape. An aneurysm is an area of local varicose veins with vessel wall thinning. The most obvious sign is the swollen veins. New Vessels Elsewhere are new vessels arising from and beyond the optic disc due to the ischemia from chronically obstructed capillaries. Optic-disc new vessels are neovascularization characterized by coils of blood vessels that develop on the optic disc or within 1 diameter of the optic disc. To distinguish optic disc neovascularization from normal small vessels, it is important to remember that normal vessels are always progressively smaller and do not rotate back towards the optic disc while neovascularization always does so, which can form a plexus in the loop with the top of the loop wider than the background. Extraoptic new vessels are located outside the optic disc

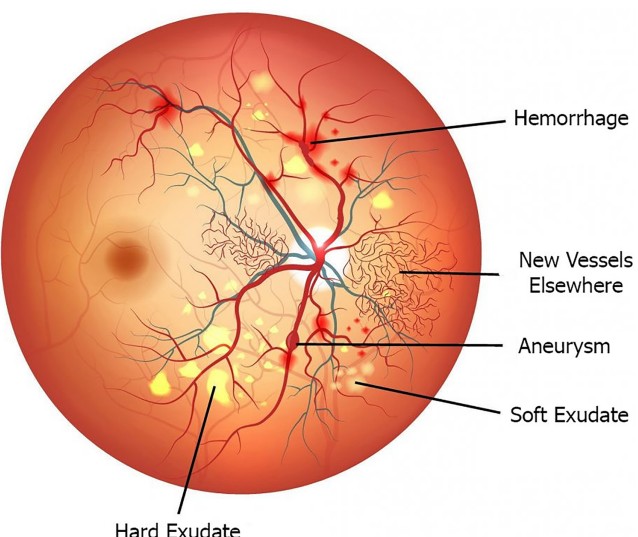

**Figure 1** **Types of lesions of diabetic retinopathy.**

neovascularization site, manifesting as a wheel-shaped network of small vessels, usually arising from the retinal veins or capillaries. Cardiac fluorescein angiography showed marked drug leakage, while retinal microvascular abnormalities showed no drug leakage. The lesion types of diabetic retinopathy are shown in Fig. 1.

In this article, we will assess the severity of the disease based on the above four types of lesions at five levels as follows: 0—No DR, 1—Mild DR, 2—Moderate DR, 3—Severe DR, and 4—Proliferate DR. In the United Kingdom, the Diabetic Eye Screening Program follows five-stage screening criteria to determine the stages of DR. Stage No DR indicates the absence of any signs of diabetic retinopathy. There are no abnormalities observed in the retinal blood vessels, and there are no clinical manifestations of the disease. In this Mild DR stage, there are minor abnormalities observed in the retinal blood vessels. These may include microaneurysms (small swellings of the blood vessels), leakage of fluid from blood vessels, and the presence of hard exudates (yellowish deposits). However, these abnormalities do not significantly affect visual acuity. The Moderate DR stage involves a further increase in abnormalities in the retinal blood vessels. The small blood vessels may become blocked or distorted, leading to the deterioration of the retinal area. Retinal exudates and hemorrhages may be present. Mild visual impairment can occur at this stage. At this Severe DR stage, there is a significant worsening of retinal deterioration. The retinal blood vessels continue to deteriorate, and there may be blockages or distortions. Visual impairment can be more pronounced at this stage. Proliferative DR is the most severe stage of diabetic retinopathy. It is characterized by the growth of abnormal blood vessels in the retina, known as neovascularization. These new blood vessels are fragile and can lead to bleeding within the retina. Fibrous membranes can form, pulling the retina away from its underlying structure and causing severe visual disturbances and even vision loss. By UK standards, two images will be required for each eye, one centered on the optic disc, and the other centered on the macular. However, in the world, people often use one image for one

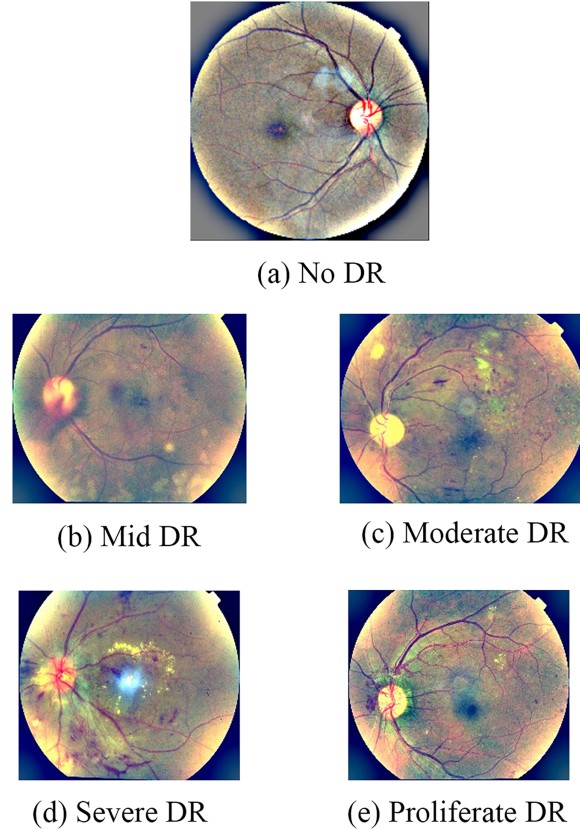

(a) No DR

(b) Mid DR          (c) Moderate DR

(d) Severe DR          (e) Proliferate DR

**Figure 2** **Five levels of DR severity: (A) No DR, (B) Mid DR, (C) moderate DR, (D) Severe DR, (E) Proliferate DR.**               

eye containing both optic disc and optic disc to assess the extent of the damage. The set of images we use will be world standard. The five severity levels of DR are shown in Fig. 2. The levels from 1 to 3 include the following types of lesions: exudative, hemorrhagic, and aneurysm. We found that at these three levels, the larger the number and area of damaged areas are, the higher the level of damage will be determined. In level 4, new proliferative blood vessels or a combination of other very severe lesions occupying a large area may appear.

A major problem with DR detection involves the difficulty of identifying symptoms in the early stages due to the similarity between images of no DR, mild DR, and sometimes moderate DR (*Porwal et al., 2020*; *Shenavarmasouleh et al., 2020*). In particular, detecting current DR requires a clinician well trained to manually assess digital color retinal images. DR is determined by locating lesions associated with diabetic vascular abnormalities. This current solution works, but it is time-consuming and highly dependent on the expertise of the trained photo reader. With the same picture, one doctor can conclude that it is a 3—DR severe state, while another doctor might think that it is only a mild 1—DR state. It is clear that differences in individual judgments are inevitable. This can cause noise during later model training and affect the results. To solve this problem, over the past few years, much research has been carried out to develop an automated solution to detect and evaluate DR

levels (*Muhammed & Toman, 2020*; *Wang & Yang, 2018*; *Trivino et al., 2018*; *Kaggle, 2019*; *Ricci & Perfetti, 2007*; *Leahy, O'Brien & Dainty, 2012*; *Shenavarmasouleh & Arabnia, 2021*).

In this article, we focus on solving two problems. First, we detect and quantify the lesion types of the disease to have a basis to assess the severity of DR, basing on the number of lesions of each type, and the ratio of each lesion type (calculated by the area of the lesion in the retinal area). To solve the problem of detecting and quantifying the lesion types of the disease, we found that the above four types of the lesion can be divided into two categories exudative lesions (hemorrhagic, exudative lesions), and vascular lesions (aneurysmal, proliferative lesions). In order to increase each type's clarity, we had to use two different image preprocessing methods, therefore we also used two sub-nets to detect the five types of lesions (hemorrhage, hard exudate, soft exudate, aneurysm, proliferation). The first sub-net was used to detect hemorrhage, hard exudate, and soft exudate while the second one was used to detect aneurysm and proliferation. Second, we provide an explanatory model to assess the severity of the disease. In the explanatory model, we use a decision tree to classify fundus images into five levels basing on features found after the quantification of lesion types.

The rest of the article is arranged as follows: the Related research section describes current studies on DR detection and assessment; the Proposed method section describes the data and methods for measuring DR severity; the Experiment and results section describes experimental results; the Conclusion section gives the conclusion.

## RELATED RESEARCH

Due to the danger of DR and the increasing number of patients, there have been many studies to solve the problem of detecting and assessing the degree of DR. These studies can be categorized as follows.

### Traditional methods

These methods usually let the computer simulate the human way, using the characteristics of each type of lesion (also called features) to automatically detect them for DR screening and classification.

Some typical vascular segmentation and detection methods (*Marín et al., 2010*; *Waly & El-Hossiny, 2020*; *Soares et al., 2006*; *Smailagic et al., 2019*; *Ronneberger, Fischer & Brox, 2015*) are outlined below. *Marín et al. (2010)*, use 7-D feature vectors and classify each pixel into two classes as vascular and non-vascular, which provides a clear picture of the vascular structure of the fundus image under different light and noise conditions. *Soares et al. (2006)* based on the pixel's feature vector, segmentations by classifying each image pixel as vessel or non-vessel. Feature vectors are two-dimensional Gabor wavelet transform responses taken at multiple scales and the pixel's intensity. The Gabor wavelet is capable of tuning to specific frequencies, allowing noise filtering and vessel enhancement in a single step. They use a Bayesian classifier with conditional probability density functions of the class described as a Gaussian mixture, making it capable of fast classifiers, and able to model complex decision surfaces. The probability distribution is estimated based on the

training set of manually labeled pixels. They used two publicly available databases, DRIVE and STARE, with manually labeled images, to evaluate method performance (*Soares et al., 2006*). *Smailagic et al. (2019)* improve the accuracy of diabetic retinopathy detection by implementing color correction and shadow removal as a pre-processing stage from the orbital image of the eye. They propose a shadow removal class that allows us to learn preprocessing functionality for a particular task (*Smailagic et al., 2019*). *Ronneberger, Fischer & Brox (2015)* presents an architecture consisting of a contracting path to capture context and a symmetric expanding path that enables precise localization. This is a network and training strategy that relies on the strong use of data augmentation to use the available annotated samples (*Ronneberger, Fischer & Brox, 2015*). *Waly & El-Hossiny (2020)* use the Gobar filter to identify blood vessels in two free retinal databases, STARE and DRIVE.

There are a number of fundus image preprocessing methods (*Foracchia, Grisan & Ruggeri, 2005*; *Leahy, O'Brien & Dainty, 2012*; *Xiong, Li & Xu, 2017*; *Cheung, Mitchell & Wong, 2010*; *Klein et al., 1984*) used to make the lesions that can be identified by the naked eye clearer, while also highlight the lesions's features in machine learning. Some prominent examples include *Smailagic et al. (2019)* using U-net architecture to create shadowless images (*Leahy, O'Brien & Dainty, 2012*); *Foracchia, Grisan & Ruggeri (2005)* normalizing the brightness and increasing image contrast by removing foreground and background pixels using a Gaussian model (*Xiong, Li & Xu, 2017*); *Leahy, O'Brien & Dainty (2012)* utilizing Laplace interpolation and a multiplicative illumination model to produce sharp images (*Cheung, Mitchell & Wong, 2010*); *Xiong, Li & Xu (2017)* proposing an image formation model related to scattering and background illumination being proposed and inverted to obtain well-illuminated images (*Klein et al., 1984*).

## Deep learning methods

Recently, convolutional neural networks (CNN) have been applied with great success in the field of computer vision. This method has shown effectiveness superior to the traditional techniques.

*Zhou et al. (2020)* establish three benchmark tasks for evaluation are DR lesion segmentation, DR grading by joint classification and segmentation, and transfer learning for ocular multi-disease identification. Moreover, a novel inductive transfer learning method is introduced for the third task. They construct a large fine-grained annotated DR dataset containing 2,842 images (FGADR) (*Zhou et al., 2020*). *Muhammed & Toman (2020)* propose work that includes visual enhancement in the visual image in the preprocessing stage, after that the CNN model is trained to be able to recognize and classify the stage, in order to diagnose the unhealthy and healthy retina image. Three public datasets DrimDB, DiaretDB0, and DiaretDB1 were used in practical testing. The authors used Matlab-R2019a, a deep learning toolbox, and a deep network designer to design and train a deep learning network (*Muhammed & Toman, 2020*). *Wang & Yang (2018)* proposes a deep-learning method to detect interpretable diabetic retinopathy. The intuitive interpretability of the proposed method is achieved by adding a regression activation map (RAM) after the global average aggregation layer of the integrated networks. With RAM,

**Table 1 Statistics of two image datasets APTOS 2019 and EyePACS.**

|  | APTOS 2019 | EyePACS | Total |
|---|---|---|---|
| Total image number | 3.662 | 35.126 | 38.788 |
| No DR | 1.805 | 25.810 | 27.615 |
| Mild DR | 370 | 2.443 | 2.813 |
| Moderate DR | 999 | 5.292 | 6.291 |
| Severe DR | 193 | 873 | 1.066 |
| Proliferate DR | 295 | 708 | 1.003 |
| Resolution | 3,216 × 2,136 | 3,888 × 2,951 |  |

the proposed model can segment the distinct regions of the retinal images to display specific regions of interest in terms of severity (*Wang & Yang, 2018*). *Toledo-Cortés et al. (2020)* propose a combined deep learning-Gaussian procedural approach to diagnose DR and quantify the uncertainty presented. This method combines the representational power of deep learning, with the ability to generalize from small datasets of Gaussian process models (*Toledo-Cortés et al., 2020*). This method combines deep learning's representative power, with the ability to generalize from small datasets of Gaussian process models. The result shows that the quantification of the prediction's uncertainty has improved the interpretability of the method as a diagnostic supporting tool. *Shenavarmasouleh et al. (2020)* build a DRDr II system based on the success of the previous version DRDr (*Shenavarmasouleh & Arabnia, 2021*). DRDr II is trained to detect and segment for the two types of exudative and microvascular lesions (hemorrhagic and proliferative). They use Kaggle's 2019 dataset with over thirty-five thousand images. The authors are able to predict the disease's severity with more than 92% accuracy.

However, these models' explanation for the assessment of disease severity is still limited, only showing lesions through images. These studies have not provided an explanation about on what basis the assessment of disease severity was based. In this article, we will focus on solving this problem.

## PROPOSED METHOD
### Data selection
In recent studies on DR, several databases have been built, such as EyePACS, APTOS 2019, MESSIDOR, DRIVE, STARE and DIARETDB. In this study, we are using two Kaggle's datasets, EyePACS (Eye Picture Archive Communication System; https://www.eyepacs.com/) and APTOS 2019 (Asia Pacific Tele-Ophthalmology Society, East Melbourne, VIC, Australia; https://www.kaggle.com/c/aptos2019-blindness-detection). These two datasets have a relatively large number of images and are already classified into the five stated degrees of severity. The statistics of the two datasets are shown in Table 1.

It can be seen that both datasets have similar resolutions, but the data distribution is severely imbalanced. There are a lot of non-DR images, while the number of heavy DR and proliferative DR images is very small. The number of severe DR and proliferate DR images each is just over 1,000 images. We therefore combine these two datasets and choose 5,000

images (1,000 images for each category). This new dataset was re-evaluated by a panel of five ophthalmologists before being put into use. Then using these 5,000 images, we cooperated with ophthalmologists to segment the lesion area to produce training data and test the model. From this dataset, 4,000 images will be selected for the training set, and 1,000 images for the testing set.

## Data preprocessing and enhancement

Before training the model, there are a few key preprocessing steps that can help the model learn better. The first is to crop the image close to the edge of the retina, removing some of the unnecessary black backgrounds that don't help train the model. The images were read and converted to grayscale to produce binary images represented by two values 0 (black) and 1 (white). Then the bounding rectangle for the object is found. The images after this step are resized to 1,024 × 1,024. At this size, the image is not too small to clearly observe the details of the damaged areas and help the model to be trained faster. The second is to increase the image contrast in order to highlight details that are hard to see with the naked eye such as blood vessels, yellow and green "mold" streaks or black spots around the retina boundary, *etc*. An example is shown in Fig. 3. It can be seen that after performing the above two preprocessing steps, the contours on the retinal image are shown more clearly. Third, because the data is not much, we apply some processing steps to data augmentation such as flipping horizontally, rotating the image, increasing the contrast and the brightness.

In addition, while conducting the experiments we found that it was difficult to identify vascular lesions (aneurysm, proliferation) since the blood vessels were often blurred and not clear, and the proliferative vessels were often very small. To overcome this problem, we used an auto-encoding model for vascular segmentation. This method was inspired by *Vincent et al. (2010)* who used auto-encoders to denoise (*Adem, 2018*) and then applied by *Fan & Mo (2016)* to segment blood vessels (*Tan et al., 2017*).

The pre-segmented image are converted to the green channel to reduce computation and help the vessels achieving high contrast (*Guo et al., 2019*). Figure 4 is an example of retinal vascular segmentation processing.

## Proposed method

Diabetic retinopathy progresses through stages from low to high severity. The assessment of these stages is currently being conducted manually through the doctors' intuition. Therefore, we propose a quantification-based lesion classifying method based on the characteristics of each type of lesion and the combination of those types of lesions together (Fig. 5).

The auto-encoding model for vascular segmentation is used to segment blood vessels in the retinal images to easily observe aneurysm or proliferative lesions.

The deep learning model for lesion detection, including two subnets. The first subnet is a mask R-CNN to detect hemorrhagic and exudative lesions, which is used to detect and segment retinal lesions such as hemorrhage and exudate and the characteristics of each lesion. The second one is a mask R-CNN to detect vascular lesions, which is used to detect

**Peer**J Computer Science

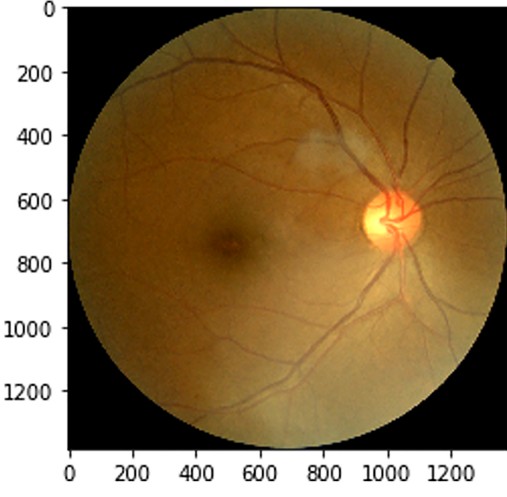

(a) Image before processing

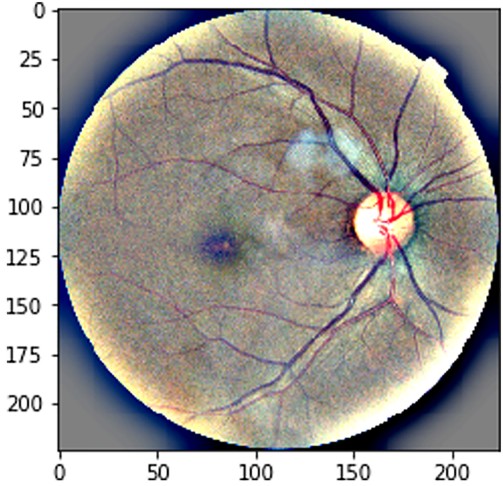

(b) Image with increased contrast

**Figure 3** Image before and after contrast enhancement: (A) Image before processing, (B) image with increased contrast.

and segment retinal lesions such as aneurysms, proliferation and the characteristics of each lesion.

An explanatory and DR lesson assessing model is used to classify and explain DR lesions.

### Auto-encoding model of vascular segmentation

Before evaluating the aneurysm and proliferative lesion, we performed vascular segmentation to produce vascular images that were clear and easy to access. There have been several studies on auto-encoding models for denoising (*Vincent et al., 2010*) and for

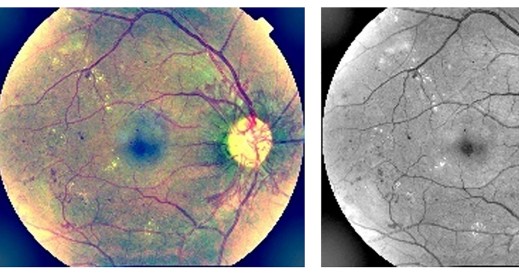

(a) Original image     (b) Green channel of the
                           original image

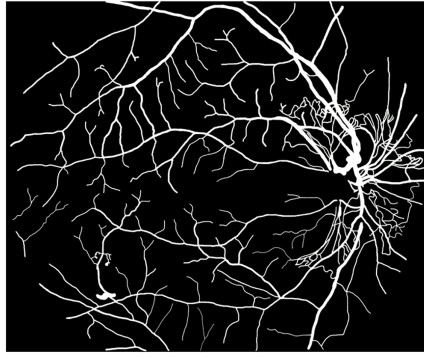

(c) Segmented image

**Figure 4** Example of retinal vascular segmentation processing: (A) Original image; (B) green channel of the original image; (C) segmented image.               

vascular segmentation (*Fan & Mo, 2016*) which have proven effective. Based on those studies, we designed the following vascular segmentation model:

Encoding parameters in vascular segmentation model training are shown in Table 2.
Decoding parameters in vascular segmentation model training are shown in Table 3.
The detailed setup parameters of the auto-encoding model are shown in Tables 2 and 3.

### Deep learning model for lesion detection

The Mask R-CNN (*Ren et al., 2015*), the deep learning model for lesion detection, is built on Faster R-CNN (*Toledo-Cortés et al., 2020*; *Abràmoff, Garvin & Sonka, 2010*). In addition to returning the label and bounding box of each object, it will also add object masks to the image.

To train the model, clinicians will have to conduct localization of lesions on the selected set of images as training data. The shape of the lesions is very diverse, it is not possible to use a rectangle to close the area to the object. Therefore we will use polygons to represent the object container. Clinicians will localize as close to the edge of the lesion as possible. In the case of small and dense lesions (usually hard exudative lesions) we will treat the whole area as one large lesion.

The model will first use ConvNet, ResNet101 architecture (Backbone), to extract features from the input image Table 4. They will be passed through a region proposal

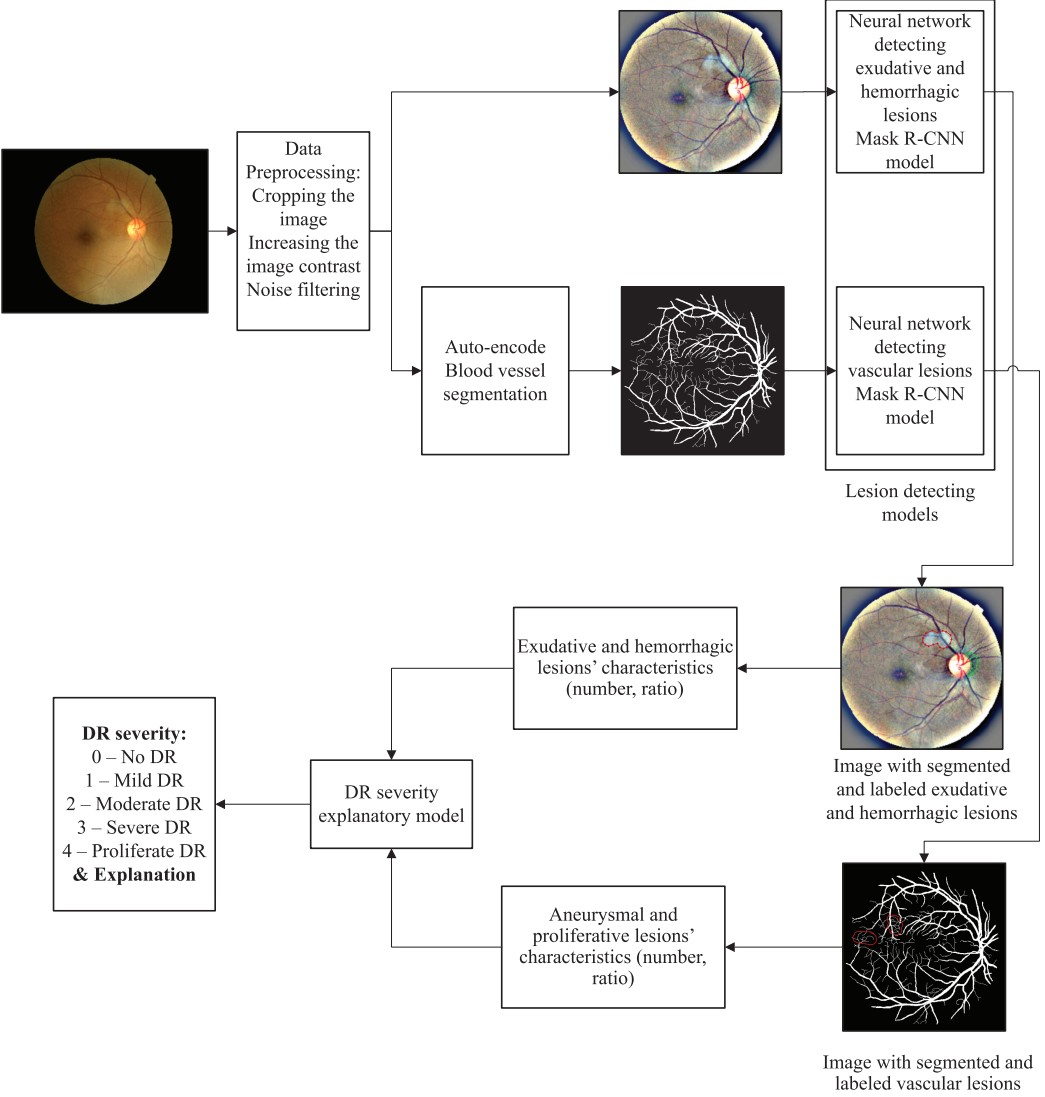

**Figure 5 Overview of the proposed method.**

network (RPN) which will then return bounding boxes of different sizes at regions that may have objects. A layer of Roi pooling layer will be added in order to aggregate all bounding boxes of the same object with different sizes to the same size. Finally, they will be passed to the fully connected layer (FC layers) branch and the mask branch for classification; and the output is a bounding box and mask for each object.

### Characteristics of lesions

From the mask object obtained after using the Mask R-CNN model, we analyzed and extracted several features that affect the assessment of lesion severity. These characteristics were calculated with each lesion type, including number of hemorrhagic lesions, ratio of hemorrhagic lesions, number of hard exudative lesions, ratio of hard exudative lesions, number of soft exudative lesions, ratio of soft exudate lesions, number of aneurysmal lesions, ratio of aneurysmal lesions, number of proliferative lesions, and ratio of

**Table 2 Encoding parameters in vascular segmentation model training.**

| Layer | Parameter | |
| --- | --- | --- |
| Conv2D | Filters | 64 |
| | Kernel_size | (3, 3) |
| | Activation | ReLU |
| | Padding | Same |
| MaxPooling2D | Pool_size | (3, 3) |
| | Padding | Same |
| Conv2D | Filters | 64 |
| | Kernel_size | (3, 3) |
| | Activation | ReLU |
| | Padding | Same |
| MaxPooling2D | Pool_size | (3, 3) |
| | Padding | Same |

**Table 3 Decoding parameters in vascular segmentation model training.**

| Layer | Parameter | |
| --- | --- | --- |
| Conv2D | Filters | 64 |
| | Kernel_size | (3, 3) |
| | Activation | ReLU |
| | Padding | Same |
| UpSampling2D | Size | (2, 2) |
| Conv2D | Filters | 64 |
| | Kernel_size | (3, 3) |
| | Activation | ReLU |
| | Padding | Same |
| UpSampling2D | Size | (2, 2) |
| Conv2D | Filters | 1 |
| | Kernel_size | (3, 3) |
| | Activation | Sigmoid |
| | Padding | Same |

**Table 4 Training parameters of the lesion detection model.**

| MASK R-CNN | | |
| --- | --- | --- |
| BACKBONE | ResNet101 | |
| FORMAT | Coco | |
| BATCH_SIZE | 4 | |
| NUM_CLASSES | Network 1 | 3 (HE, EX, SE) |
| | Network 2 | 2 (AN, NV) |
| AUGMENTATION | TRUE | |
| NUM_EPOCHS | 300 | |

**Table 5 Some terms used**

| Terms | Describe |
|---|---|
| $t$ | Type of lesion such as hemorrhagic plaque, soft exudate, hard exudate, aneurysm, proliferative |
| $n^t$ | The number of lesions of each type are integers greater than or equal to 0 |
| $T^t$ | Total injury rate $t$ |
| $T_i^t$ | Ratio of the $i$th lesion object of type $t$ |
| $S$ | Area of retina |
| $S_i^t$ | Lesion area of the $i$th lesion object, type $t$ |

proliferative lesions. As for aneurysmal lesion, we were only interested in the number of lesions, not the ratio, since this type of lesion has very small area of damage. Specifically, the results after running the lesion detection model on a fundus image will obtain a zoned image labeled with the lesion and the array of the segmentation's output. The array of the segmentation's output includes detected objects' arrays, objects' corresponding class_ids' arrays, segmentation masks' arrays, and the output's array. From the class_ids' array of the subjects, we counted the number of each lesion type. From the segmentation masks array, we get the list of point coordinates that make up the image mask to calculate the mask area. Table 5 is some of the terms used.

Ratio of lesions for each type is calculated by dividing the total area of the lesion of that type to the retinal area.

$$T^t = \sum_{i=0}^{n^t} T_i^t \tag{1}$$

where:

$$T_i^t = S_i^t / S \tag{2}$$

where:

$$S_i^t = \left| \sum (x_j * y_{j+1}) - (y_j * x_{j+1}) \right| / 2 \tag{3}$$

Each object's area of damage is calculated using vertices coordinates based on a list of coordinates of the polygon's vertices. With $(x_j, y_j)$ is the coordinates of vertex $j$, and the last vertex will be connected to the first vertex to create a polygon $x_{n+1} \rightarrow x_1, y_{n+1} \rightarrow y_1$.

In reality, the calculation of the retinal area has problems with images taken as follows.

(1) There are images that are missing the upper or lower parts like Fig. 6.

In order to solve this problem, we considered the retina to be a perfect circle, and the retinal area is equal to the area of the circle surrounding it $S = R^2 * \pi$, where R is the radius of the circle, which is also equal to 1/2 the width of image after being trimmed in the preprocessing step.

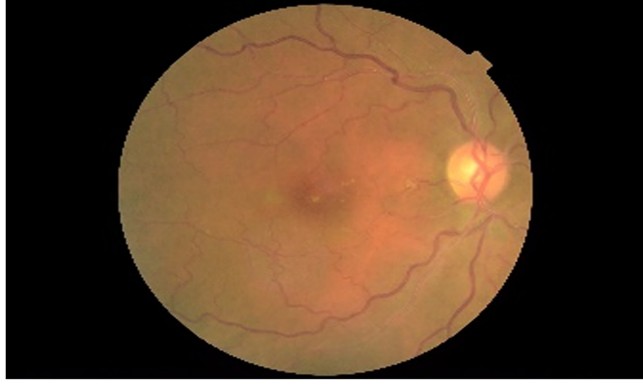

(a) Full image

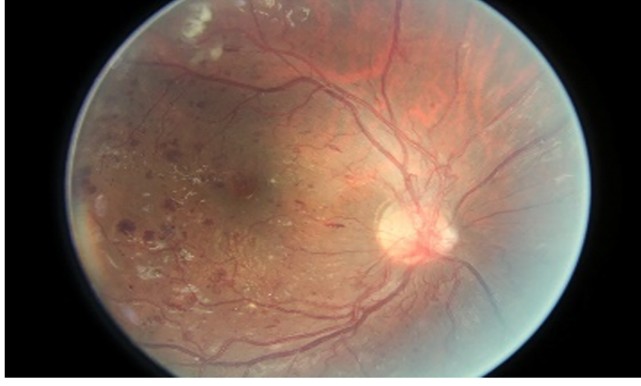

(b) Image with missing parts

**Figure 6** **Problem with retinal images: (A) Full image; (B) image with missing parts.**

(2) There are images with different resolution. Because the dataset is merged from two different database, the images have different resolution. Therefore, we will not use directly the value of the image area, instead we will convert them to the ratio of lesion object.

### Vulnerability assessment model of DR & explanation

For the convenience of presentation, the following in Table 6 abbreviations will be used in the article.

In this article, we use the decision tree to evaluate and interpret the lesion severity, based on nine properties including four lesion ratios R_HE, R_EX, R_SE, R_NV, and five lesion numbers N_HE, N_EX, N_SE, N_AN, N_NV to classify into five categories 0—No DR, 1—Mild DR, 2—Moderate DR, 3—Severe DR, 4—Proliferate DR. Which, the values of the lesion attributes are calculated as follows:

$$\text{R\_HE} = T^{HE} = \sum_{i=0}^{n^{HE}} T_i^{HE} = \sum_{i=0}^{n^{HE}} S_i^{HE}/S$$
$$\text{R\_EX} = T^{EX} = \sum_{i=0}^{n^{EX}} T_i^{EX} = \sum_{i=0}^{n^{EX}} S_i^{EX}/S$$

**Table 6 Abbreviations used in the article.**

| Acronyms | Explain | Acronyms | Explain |
|----------|---------|----------|---------|
| EXS | Exudate | N_EX | Number of hard exudative lesions |
| EX | Hard exudates | R_EX | Ratio of hard exudative lesions |
| SE | Soft exudate | N_SE | Number of soft exudative lesions |
| HE | Hemorrhage | R_SE | Ratio of soft exudate lesions |
| AN | Aneurysm | N_AN | Number of aneurysmal lesions |
| NV | New vessels elsewhere | R_AN | Ratio of aneurysmal lesions |
| N_HE | Number of hemorrhagic lesions | N_NV | Number of proliferative lesions |
| R_HE | Ratio of hemorrhagic lesions | R_NV | Ratio of proliferative lesions |

$$\text{R\_SE} = T^{SE} = \sum_{i=0}^{n^{SE}} T_i^{SE} = \sum_{i=0}^{n^{SE}} S_i^{SE}/S$$

$$\text{R\_NV} = T^{NV} = \sum_{i=0}^{n^{NV}} T_i^{NV} = \sum_{i=0}^{n^{NV}} S_i^{NV}/S$$

$$\text{N\_HE} = n^{HE}$$

$$\text{N\_EX} = n^{EX}$$

$$\text{N\_SE} = n^{SE}$$

$$\text{N\_AN} = n^{AN}$$

$$\text{N\_NV} = n^{NV}$$

In order to evaluate and interpret the DR severity, we use decision tree because of its ease of understanding, without the need for additional tools and considerable computing power (*La Malfa et al., 2021*). First, decision trees has a graphical structure. Second, decision trees often contain a subset of attributes, helping the users to focus on analyzing the most relevant ones. Third, the hierarchical tree structure provides information about the relative importance of different attributes (*Freitas, 2014*). We used C&R Tree, CHAID, Tree-AS, Quest algorithms on the dataset of 5,000 images. Among which, the C&R Tree model produced the most accurate results, and the results are shown in Table 7. Therefore, for the explanatory model, we used the Decision Tree-Classification and Regression Trees (CART) algorithm.

After using the CART Decision Tree, we received the following rules shown in Fig. 7. For convenience, the rules found are summarized in Table 8. For example, the rule that determine that the severity degree is Moderate DR R_SE=<0.18 & R_HE>0.052 & R_HE=<0.171 can be interpreted as: If the ratio of soft exudate lesions is less than or equal 18% and the ratio of hemorrhage lesions is greater than 5.2% and the ratio of hemorrhage lesions is less than or equal 17.1%, the severity is moderate.

## EXPERIMENT AND RESULTS

The experiments reported in this article were run on a computer equipped with an NVIDIA Quadro GTX 1080 GPU. The computer has an Intel® Core™ i7 processor with four cores, eight 2.2 GHz threads, and 32 GB of RAM. The software packages used to

**Table 7  The accuracy of the explanatory models.**

|   | Model | Accuracy (%) | Number of fields used |
|---|-------|--------------|-----------------------|
| 1 | C&R Tree | 96.24 | 8 |
| 2 | Tree-AS | 92.18 | 6 |
| 3 | CHAID | 92.10 | 5 |
| 4 | Quest | 87.44 | 8 |

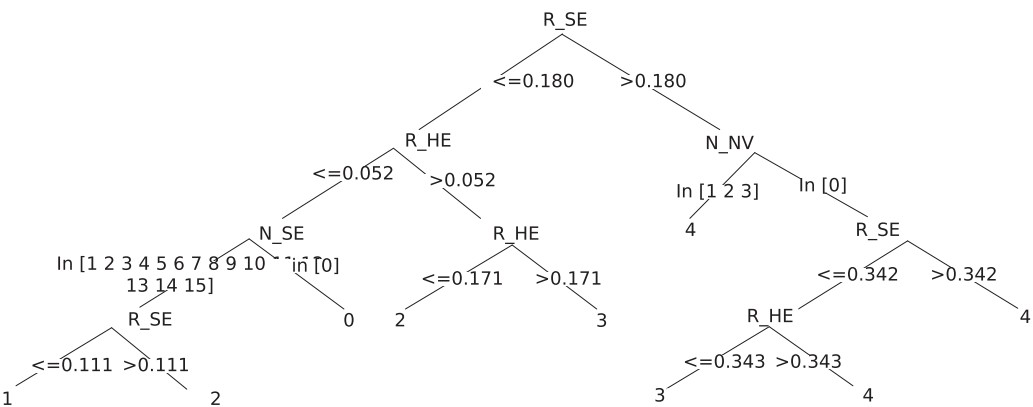

**Figure 7  Decision tree to determine the extent of damage to the fundus image.**

**Table 8  Several rules determining DR severity.**

| DR severity | Rule | Rule confidence (%) |
|-------------|------|---------------------|
| No DR | R_SE=<0.18 & R_HE=<0.052 & N_SE=0 | 95.125 |
| Mild DR | R_SE=<0.18 & R_HE=<0.052 & N_SE=[1, …, 15] & R_SE=<0.111 | 99.396 |
| Moderate DR | R_SE=<0.18 & R_HE=<0.052 & N_SE=[1, …, 15] & R_SE>0.111 | 95.238 |
|  | R_SE=<0.18 & R_HE>0.052 & R_HE=<0.171 | 99.700 |
| Severe DR | R_SE=<0.18 & R_HE>=0.052 & R_HE>0.171 | 58.824 |
|  | R_SE>0.18 & N_NV=0 & R_SE=<0.342 & R_HE=<0.343 | 92.381 |
| Proliferate DR | R_SE>0.18 & N_NV=[1, 2, 3] | 100.000 |
|  | R_SE>0.18 & N_NV=0 & R_SE>0.342 | 100.000 |
|  | R_SE>0.18 & N_NV=0 & R_SE=<0.342 & R_HE>0.343 | 100.000 |

**Table 9  Calculated scores for each class.**

| Lesion type | Score | | | |
|-------------|-----------|--------|------|----------|
|  | Precision | Recall | F1 | Accuracy |
| HM | 0.96 | 0.98 | 0.97 | 0.94 |
| HE | 0.98 | 0.95 | 0.96 | 0.93 |
| SE | 0.97 | 0.95 | 0.96 | 0.92 |
| AN | 0.97 | 0.98 | 0.97 | 0.95 |
| NV | 0.97 | 1 | 0.98 | 0.97 |
| Average: |  |  |  | 0.942 |

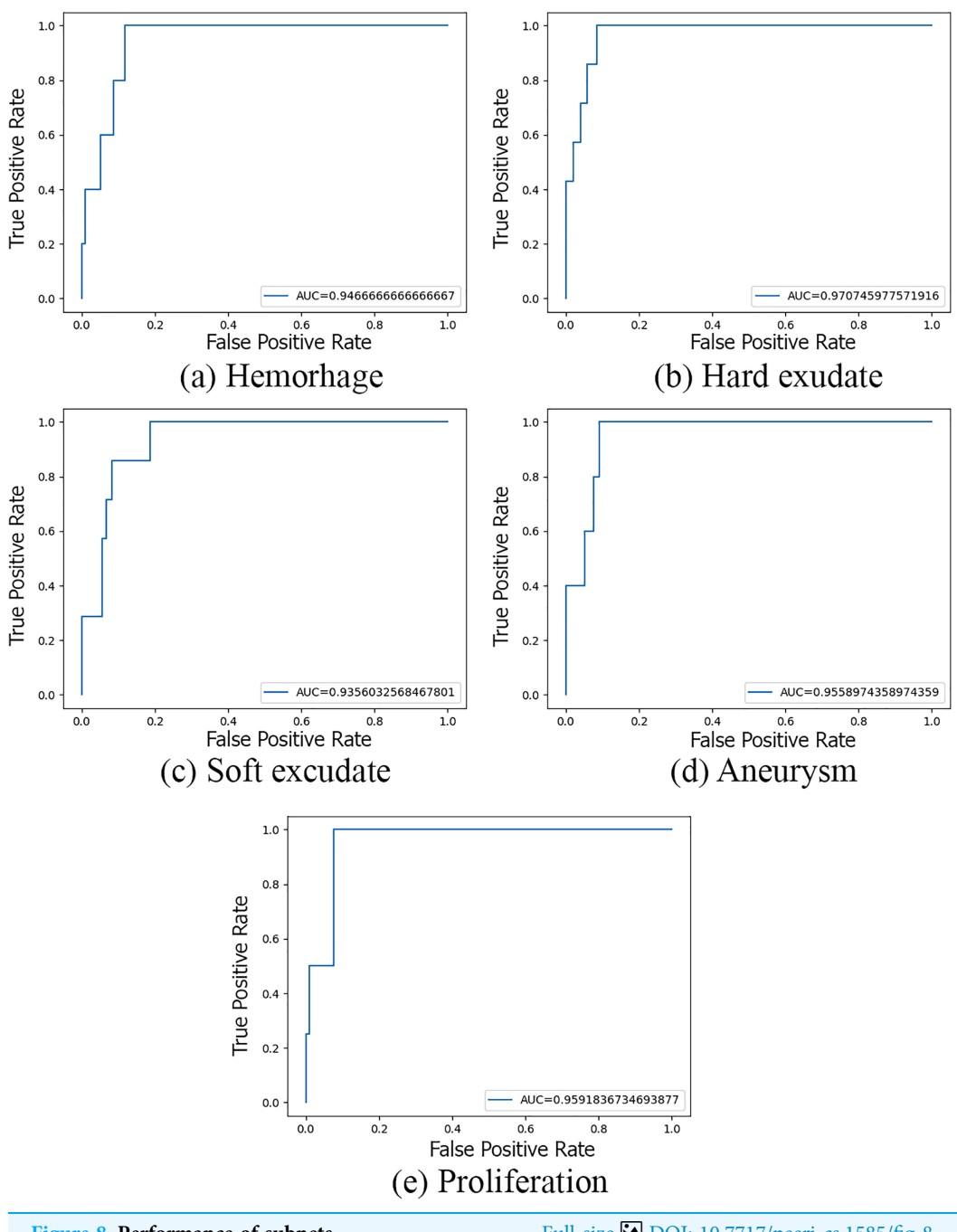

**Figure 8  Performance of subnets.**     

deploy the models are Python 3.8 along with deep learning libraries like Keras, Tensorflow, OpenCV, Pixellib.

Because the number of lesions of each lesion type are different on each image, there are common lesion types such as hemorrhage, hard exudate, soft exudate while the lesion types like aneurysm and proliferation are rarer. Therefore, in order to avoid skewed data, we evaluated each lesion type separately. For each lesion type, we randomly selected from

**Table 10 Comparison with other lesion detecting methods.**

| Reference | Task | Dataset | AUC | Sensitivity (SE) | Specificity (SP) |
|---|---|---|---|---|---|
| *Van Grinsven et al. (2016)* | HE | Kaggle | 89.4 | 83.7 | 85.1 |
| *Van Grinsven et al. (2016)* | HE | Messidor | 97.2 | 91.9 | 91.4 |
| *Adem (2018)* | Exudates | DiaretDB0 | – | 100 | 98.41 |
| *Adem (2018)* | Exudates | DrimDB | – | 100 | 98.44 |
| *Tan et al. (2017)* | EX | Cleopatra | – | 87.58 | 98.73 |
| *Tan et al. (2017)* | HE | Cleopatra | – | 62.57 | 98.93 |
| *Tan et al. (2017)* | MA | Cleopatra | – | 46.06 | 97.99 |
| *Guo et al. (2019)* | EX | DDR | 55.46 | – | – |
| *Guo et al. (2019)* | SE | DDR | 26.48 | – | – |
| *Guo et al. (2019)* | HE | DDR | 35.86 | – | – |
| *Guo et al. (2019)* | MA | DDR | 10.52 | – | – |
| *Abràmoff et al. (2010)* | MA, HE | – | – | 47.7 | 90 |
| *Giancardo et al. (2011)* | MA | – | – | 50 | >10 false positive per image |
| *Niemeijer et al. (2009)* | MA | – | – | 60 | 8 false positive per image |
| *Mizutani et al. (2009)* | MA | – | – | 65 | 27 false positive per image |
| *Quellec et al. (2008)* | MA | – | – | 90.24 | 89.75 |
| *Walter et al. (2007)* | MA | – | – | 88.50 | 2.13 false positive per image |
| *Huang, Yan & Aviyente (2007)* | MA | – | – | 68 | >40 false positive per image |
| *Niemeijer et al. (2007)* | Exudates | – | – | 95 | 88 |
| *Philip et al. (2007)* | MA, HE | – | – | 97.90 | 67.40 |
| *Fleming et al. (2006)* | MA | – | – | 85.40 | 83.10 |
| *Quellec et al. (2006)* | MA | – | – | 87.90 | 96.20 |
| *Pallawala et al. (2005)* | MA | – | – | 93 | NA |
| *Serrano, Acha & Revuelto (2004)* | MA | – | – | 90.72 | 82.35 |
| *Niemeijer et al. (2005)* | MA, HE | – | – | 100 | 87 |
| *Larsen et al. (2003)* | MA, HE | – | – | 96.70 | 71.40 |
| *Rapantzikos, Zervakis & Balas (2003)* | Drusem | – | – | 98.80 | 99.31 |
| *Sinthanayothin et al. (2002)* | MA, HE | – | – | 77.50 | 88.70 |
| *Hsu et al. (2001)* | EXS | – | – | 100 | 74 |
| *Yang et al. (2001)* | MA | – | – | 90 | 80 |
| *Wang et al. (2000)* | EXS | – | – | 100 | 70 |
| *Hipwell et al. (2000)* | MA | – | – | 85 | 76 |
| *Ege et al. (2000)* | MA, HE, EXS, SE | – | – | 94 | 69 |
| *Lee, Wang & Lee (1999)* | MA, HE | – | – | – | – |
| *Cree et al. (1997)* | MA | – | – | 82 | 84 |
| *Gardner et al. (1996)* | HE, EXS | – | – | 73.80 | 73.80 |
| *Spencer et al. (1992)* | MA | – | – | 45 | >150 false positive per image |
| Our method | HE | EyePACS & APTOS 2019 | 94.66 | 97.95 | 60 |
| Our method | EX | EyePACS & APTOS 2019 | 97.07 | 94.89 | 71.42 |
| Our method | SE | EyePACS & APTOS 2019 | 93.56 | 94.84 | 66.66 |
| Our method | AN | EyePACS & APTOS 2019 | 95.58 | 97.93 | 62.5 |
| Our method | NV | EyePACS & APTOS 2019 | 95.91 | 100 | 50 |

**Table 11 The results of the measurements of the model.**

| DR severity | Score | | | |
|---|---|---|---|---|
| | Precision | Recall | F1 | Accuracy |
| No DR | 0.98 | 0.98 | 0.98 | 0.97 |
| Mild DR | 0.92 | 0.88 | 0.90 | 0.81 |
| Moderate DR | 0.96 | 0.88 | 0.92 | 0.86 |
| Severe DR | 0.95 | 0.95 | 0.95 | 0.91 |
| Proliferate DR | 1.00 | 0.95 | 0.98 | 0.95 |
| Avg | | | | 0.90 |

two datasets APTOS 2019 and EyePACS so that the number of each lesion type is exactly 100.

We calculated the Precision, Recall, F1, Accuracy scores for each class and the results are shown in Table 9:

The performance of the lesion detecting subnets is shown in Fig. 8.

According to the obtained results, the AUC scores of the two subnets fluctuated from 0.93 to 0.97. In which, the detection of hard exudative lesions has the highest accuracy while the detection of soft exudative lesions has the lowest. We have compared our model with many existing lesion detecting methods, and it shows that ours is adequate. The results are listed in Table 10.

To evaluate the accuracy of the whole model, we built a 1,000-image dataset consisting of 200 images for each lesion level. These images are not included in the training dataset. We calculate the measure for each class, the results are shown in Table 11.

The system's average accuracy is 90%. We think that with such accuracy it can be a reference source for the doctor's conclusions in practice.

## CONCLUSION

We have presented a simple but effective approach to detect and create segmentation masks for five types of lesions (EX, SE, HE, AN, NV) caused by diabetic retinopathy. In the context of limited hardware resources in our research system, our study demonstrates the feasibility of our approach. The process of delineating and labeling data for retinal fundus images requires significant effort due to the diverse and complex nature of the lesion areas. Training such a solution is also more complex and time-consuming compared to classical methods. Despite the hardware limitations, we have successfully developed an effective method for detecting and creating segmentation masks for various types of ocular lesions. However, to deploy this system in practical settings, a higher-configured machine system is required, along with a substantial investment of resources in data labeling. At the same time, unlike the previous models, our model is capable of explaining its conclusion of lesion severity assessment basing on the characteristics of the number and ratio of each retinal lesion type. These explanations are very close to the doctor's clinical practice, based on the greater the number of lesions or incidence of lesions, the severity of the lesions.

Doctors and patients can visually observe the marked localized lesions and have explanations for assessing the extent of the lesion. Therefore, it has the potential to be applied easily in healthcare facilities.

### Funding

This work was supported by the CS'21.07 project "Research and development of a method to detect and evaluate retinal damage due to diabetes complications", Institute of Information Technology, Vietnam Academy of Science and Technology. The funders had no role in study design, data collection and analysis, decision to publish, or preparation of the manuscript.

### Grant Disclosures

The following grant information was disclosed by the authors:
CS'21.07 Project "Research and Development of a Method to Detect and Evaluate Retinal Damage due to Diabetes Complications", Institute of Information Technology, Vietnam Academy of Science and Technology.

### Competing Interests

The authors declare that they have no competing interests.

### Author Contributions

- Quang Toan Dao conceived and designed the experiments, performed the experiments, analyzed the data, prepared figures and/or tables, authored or reviewed drafts of the article, and approved the final draft.
- Hoang Quan Trinh performed the experiments, performed the computation work, prepared figures and/or tables, and approved the final draft.
- Viet Anh Nguyen conceived and designed the experiments, authored or reviewed drafts of the article, and approved the final draft.

### Data Availability

    The data is available at Kaggle's APTOS 2019 Blindness Detection: https://www.kaggle.com/c/aptos2019-blindness-detection.
    The data is available at Kaggle DR dataset (EyePACS):
https://www.kaggle.com/datasets/mariaherrerot/eyepacspreprocess.

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
