# Peer review of "An effective and comprehensible method to detect and evaluate retinal damage due to diabetes complications"

_PeerJ Computer Science, doi:10.7717/peerj-cs.1585_

## Round 0.1 · original submission · Major Revisions

Please consider the comments to improve the quality of the submission.

Reviewer 1 ·

Basic reporting

1. The paper presents a different, accurate and safe (from the point of view of the ophthalmologist's understanding) approach to the diagnosis of diabetic retinopathy.
2. Well-described motivation and need to create such an approach.
3. A good and insightful description of the types of approaches in automated DR diagnostic support.
4. In my opinion, the language level is good.

Experimental design

The proposed approach is interesting enough to encourage me to reproduce it. That being said, there were a few concerns that are worth addressing in the description.

2.1 Data Selection:
- 1000 images in each decision class is a bit low, perhaps it is worth explaining the reasons? This could have been supplemented with other datasets, but the problem perhaps lay in the "feasibility" of specialists' descriptions of these images?
- Perhaps it is worth adding a small addition of exactly how many photos went into the training collection (4,000) and how many went into the test collection (1,000)?
- Were each of the images marked by one ophthalmologist, or by several? Was a "ground truth" designated in such a situation? This affects the level of confidence for the data.
- Is the number of patients in both datasets known? If so, it is worth indicating this by also adding how many images per patient are in the collection.

2.2 Data Preprocessing and Enhancement:
- Line 157: „The images are read and converted to grayscale to produce binary images with a threshold value of 1”. The term "threshold value of 1" in this sentence is vague. It is worth clarifying.
- Line 161: „The third is applying some processing steps to enhance the data, such as flipping horizontally and vertically, rotating the image, increasing the contrast and the brightness.”. What do these additional operations give us? Do they involve data augumentation in order, for example, to avoid overfitting by the model? It might be worth describing.

2.3 Proposed Method
- A good diagram (figure 5) describing the general flow

2.3.1
- Figure 6 is rather redundant. It is a repetition of tables 2 and 3
- It is worth adding information about the input size of images for each model (in the appropriate places)

2.3.2
- Table 4 indicates that Transfer Learning was used, based on parameters trained on the "Coco" collection. Perhaps it is worth pointing this out openly (if it was)?
- It is also worth adding more information about the training process: optimizer, lr, lr scheduling, weight initializers, regularizers, methods of selecting hyperparameters. This applies to each model. If they come from a reference work, add such a note. If a "higher level" framework than TensorFlow or PyTorch, strictly specialized for very complex network architectures, was used for training, it is also worth pointing this out.

2.3.4
- Decision tree is a good choice when the model is to be easily explained and interpreted
- Figure 8 AND table 8 very accurately show the main idea behind the novelty of the final classifier

Validity of the findings

3. Results
- It is good that the hardware and software used in the experiment was described
- If there is a possibility, it is worth adding information about the training time of individual models
- Accuracy due to balanced decision classes I guess that it is global. Maybe it is worth pointing it out openly?
- Clinical problems also use such metrics as specificity, sensitivity, (optionally also NPV and PPV). It is worth pointing out these measures for the final model. This makes sense especially when analyzing Table 10.

4. Conclusions
- The final system consists of multiple models, which may require more computational capacity (than classical approaches) in the prediction phase. This is worth mentioning.
- Training such a solution can be more complicated and time-consuming than classical approaches.
- The final model (decision tree) is well chosen and explained without the need for additional tools and considerable computing power. It is worth referring to other solutions (e.g. neural network explanation: Emanuele La Malfa, Rhiannon Michelmore, Agnieszka M. Zbrzezny, Nicola Paoletti, Marta Kwiatkowska: On Guaranteed Optimal Robust Explanations for NLP Models. IJCAI 2021: 2658-2665) which are much more complicated and expensive in terms of computational and time complexity.

Additional comments

No

·

Basic reporting

1. Better separation of citation and the sentence content is needed to improve clarity
2. This sentence needs correction. "Background retinopathy has forms of the retinal capillary 35 aneurysm, slight bleeding, stagnation of secretions in the retina, and retinal edema - "
3. In background, a better connection between diabetes and retinopathy would be great. Some connection was drawn in the second paragraph but it's not clear how diabetes cause retinopathy.
4. Figures needs to be put in order.

Experimental design

1. I'm not sure of the criteria the clinician decided DR level for the dataset.
2. It seems the dataset is subsampled. Is it random?
3. It seems some segmention is done by clinicians. More details are needed on this. What has been done and what's the criteria?
4. Please distinguish preprocessing and the augmentation steps.
5. Please share code on Github and notebooks which others can reproduce.

Validity of the findings

1. Please give more figures of lesion segmentation results like the right bottom of Figure 5. Ideally, it should cover all type of lesions and some mixed ones.
2. The authors are firstly doing a segmenation job. Metrics for segmention is also needed in addition to classificaiton. Please share AP values.
3. Ideally, an independent test set is needed to evaluate the performance. Data from a third source or collected locally would all work.

Reviewer 3 ·

Basic reporting

The proposed model is hardly mentioned in the summary. In the summary, it should be mentioned why the study was conducted, how the proposed model solved these problems and the results obtained.

Experimental design

Unfortunately, only a table and AUC curves are presented in the experimental results section of the article. The proposed model can be compared with different models accepted in the literature. Confusion matrix etc. can be presented. It is possible to write more fluently.

Validity of the findings

In the Introduction section, the novelty of the article and its contribution to the literature should be explained. In the relevant studies section, the method used by the researchers, the results they obtained, and the lack of the study should be mentioned. Reference should be made to the equations used in line 256. The experimental result part of the work should be rewritten. If possible, this section is confusion matrix etc. should be supported. Unfortunately, a table and AUC curves are given in this section. Table 10 should be presented in the discussion section. A section called Discussion should be added to the study, and limitations of the study should be discussed in this section. The article should be reorganized. Unfortunately, it does not seem possible for me to accept your article as it is.

Additional comments

Please review specifically the organization and experimental results section of the article.

·

Basic reporting

Minor typographical issues ex. Line 35, 180, defining full forms of abbreviations when they appear for the first time in a manuscript. Ex. Line 194 and maybe usage of parenthetical citation for clear separation between manuscript and citation. Otherwise the manuscript is appropriate.

Experimental design

The method uses 5K images for training with a split of 1000 images for each category. I do see a reference to usage of enhancing the data by flipping the images but a small dataset comes with its own challenges like managing outliers that deviate from the rest of dataset, overfitting, etc. It would be a great to see how these challenges were addressed while model training.

Validity of the findings

The method is highly robust and so is the data presented for the model with its accuracy!

Additional comments

I agree with the authors assessment on its applications in hospital facilities. This model can be run every time a patient visits for an eye exam which can help the doctors for diagnosis. With an appropriate patient consent, the images captured in the patient visits can be used to retrain the model for supporting diverse data to improve the model.

---

## Round 0.2 · accepted · Accept

Thank you for addressing the reviewer's comments, your manuscript is ready for publication.

Reviewer 1 ·

Basic reporting

The authors have resolved all the questionable issues that I pointed out in my review of the original version of the paper.

Experimental design

Correct

Validity of the findings

Correct

Additional comments

After revisions, the work presents a high scientific value and will certainly leave a lasting mark in the literature. I wish the authors continued success.